# Reversible long-range domain wall motion in an improper ferroelectric

Manuel Zahn [1,2], Aaron Merlin Müller [3], Kyle P. Kelley [4], Sabine Neumayer [4], Sergei V. Kalinin [5], István Kézsmarki [2], Manfred Fiebig [3], Thomas Lottermoser [3], Neus Domingo [4], Dennis Meier [1] ✉ & Jan Schultheiß [1,6] ✉

Reversible ferroelectric domain wall movements beyond the 10 nm range associated with Rayleigh behavior are usually restricted to specific defect-engineered systems. Here, we demonstrate that such long-range movements naturally occur in the improper ferroelectric $ErMnO_3$ during electric-field-cycling. We study the electric-field-driven motion of domain walls, showing that they readily return to their initial position after having traveled distances exceeding 250 nm. By applying switching spectroscopy band-excitation piezoresponse force microscopy, we track the domain wall movement with nanometric spatial precision and analyze the local switching behavior. Phase field simulations show that the reversible long-range motion is intrinsic to the hexagonal manganites, linking it to their improper ferroelectricity and topologically protected structural vortex lines, which serve as anchor point for the ferroelectric domain walls. Our results give new insight into the local dynamics of domain walls in improper ferroelectrics and demonstrate the possibility to reversibly displace domain walls over much larger distances than commonly expected for ferroelectric systems in their pristine state, ensuring predictable device behavior for applications such as tunable capacitors or sensors.

Ferroelectrics have a switchable spontaneous polarization which is used, for example, in memory applications[1], and their large piezoelectric and dielectric responses are essential for capacitors, sensors, and actuators[2,3]. Proper ferroelectrics, whose primary order parameter is the spontaneous polarization, such as $Pb(Zr,Ti)O_3$ and $BaTiO_3$, exhibit a qualitatively well-defined electric-field-driven polarization reversal process[4] that is determined by the elastic and electrostatic boundary conditions[5,6]. At the local scale, the reversal is governed by domains, following material-dependent nucleation and growth processes that involve the movement of ferroelectric domain walls[7]. In general, for small electric fields reversible domain wall movements can occur (Rayleigh's law), whereas larger electric fields lead to nonlinearity, manifesting as Barkhausen jumps[8]. Characteristic distances reported for reversible ferroelectric domain wall movements according to the Rayleigh's law are usually in the sub-10-nm range[9]. To increase the regime in which reversible domain wall movements occur, different approaches have been tested, including point defect[10–12] and domain engineering[13], as well as cycling of ferroelectric domain walls between pinning centers[14]. Following these approaches, electric-field-driven reversible domain wall motions over distances up to several micrometers have been achieved. In contrast to proper ferroelectrics, the local dynamics and reversible displacements of domain walls in

[1]Department of Materials Science and Engineering, Norwegian University of Science and Technology (NTNU), Trondheim, Norway. [2]Experimental Physics V, Center for Electronic Correlations and Magnetism, University of Augsburg, Augsburg, Germany. [3]Department of Materials, ETH Zurich, Zurich, Switzerland. [4]Center for Nanophase Materials Science, Oak Ridge National Laboratory, Oak Ridge, TN, USA. [5]Department of Materials Science and Engineering, University of Tennessee, Knoxville, TN, USA. [6]Department of Mechanical Engineering, University of Canterbury, Christchurch, New Zealand. ✉e-mail: dennis.meier@ntnu.no; jan.schultheiss@ntnu.no

improper ferroelectrics, where the electric polarization arises as a secondary order parameter[15,16], are relatively unexplored.

Here, we investigate the domain wall dynamics in the improper ferroelectric model system ErMnO₃. By applying band-excitation piezoresponse force microscopy (BE-PFM)[17], we map the ferroelectric domains and resolve their local response to applied electric fields. We find that away from the topologically protected structural vortex lines, which are characteristic for the domain pattern in hexagonal manganites[18–20], domain walls readily move under the electric field. Most intriguingly, the domain walls consistently return to their original positions after bipolar electric field cycling, even after being displaced by distances exceeding 250 nm. Complementary phase field simulations exhibit the same switching behavior, indicating that this phenomenon is characteristic for hexagonal manganites and linked to their improper ferroelectric order.

## Results

ErMnO₃ is a geometrically driven improper ferroelectric, where the primary order parameter relates to a structural trimerization mode. Its fundamental domain[19,20] and domain-wall physics[21,22] are well-understood, making it an ideal model system for dynamical investigations at the nanoscale. The system is a uniaxial ferroelectric with a spontaneous polarization, $P \approx 6\,\mu C/cm^2$, parallel to the hexagonal $c$-axis (space group $P_6 3cm$). The domain structure consists of ferroelectric 180° domains that meet at topologically protected six-fold structural vortex lines, which from at the improper ferroelectric phase transition as described elsewhere[23]. The family of hexagonal manganites is attracting broad attention due to its unusual physical properties, including multiferroicity[24,25], functional domain walls[22,26], Kibble-Zurek scaling[27], and inverted domain-size / grain size scaling[28,29], with potential applications in barrier layer capacitors[30]. In this work, we use polycrystalline ErMnO₃ from the same batch as studied in ref. 28, where the interested reader can find relevant details concerning the processing conditions of the sample and a crystallographic and microstructural characterization.

We begin by measuring the local electric-field response using BE-PFM[17,31] as presented in Fig. 1. In contrast to conventional PFM, which operates at a single frequency, the response is detected in BE-PFM within a specified frequency range (here 360 to 420 kHz), allowing to capture the broad mechanical resonance frequency of the tip-sample system[32]. Fig. 1a shows a schematic illustration of the measurement principle. To image the polarization-reversal process, a bipolar triangular voltage signal sequence (Fig. 1a) regularly used in SS-PFM[33], is applied point by point to the surface via the probe tip. The BE-PFM response is measured after pulse application, as shown in the inset to Fig. 1a (the protocol applied to correct for electrostatic background contributions and electric-field screening effects[34,35] is described in Supplementary Figs. S1 and S2). The initial vertical BE-PFM phase response of our polycrystalline ErMnO₃ specimen is displayed in Fig. 1b (for corresponding amplitude data see Supplementary Fig. S3). We find that the BE-PFM response is maximized in the vertical channel, which leads us to the conclusion that the grain has primarily an out-of-plane polarization component, i.e., the $c$-axis is approximately perpendicular to the imaged surface. Ferroelectric $\pm P$ domains can be distinguished based on their characteristic BE-PFM phase response, with $-P$ domains showing a phase shift of 0 rad and $+P$ domains showing a $-\pi$ rad phase shift, consistent with their opposite polarization directions. The BE-PFM data shows the well-established domain structure of ErMnO₃[18,19,22], consisting of ferroelectric 180° domains that come together by forming a characteristic sixfold line defect. The surface intersection point of the line is marked by the red arrow in Fig. 1b.

Representative BE-PFM phase images recorded at maximum negative (−12.0 V) and positive (+12.0 V) voltages are displayed in Fig. 1b and c, respectively. We find pronounced spatial variations in the response of the ferroelectric domains, depending on the local domain

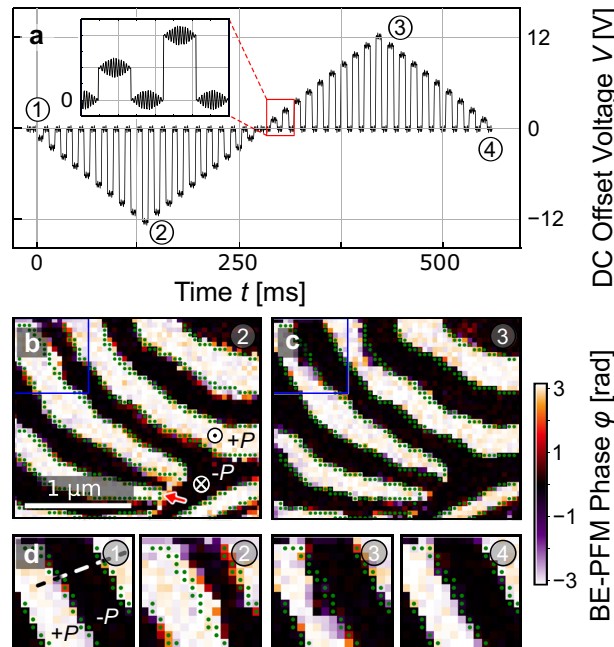

**Fig. 1 | BE-PFM data showing the voltage-dependent domain-structure evolution of ErMnO₃. a** Schematic time-dependence of the applied DC bipolar voltage sequence, corresponding to a square triangular function. The inset schematically visualizes the BE-PFM measurement applied to monitor the piezoresponse around the tip-sample resonance frequency. Details are provided in the methods section. Corrected off-field BE-PFM phase images of the domain structure at **b** maximum negative and **c** maximum positive voltage. White areas represent domains with upward polarization, $+P$, whereas black areas indicate downward polarization, $-P$. The red arrow indicates the position of the intersection of a vortex line with the surface. Green dots indicate the position of the ferroelectric domain wall prior to the application of the bipolar volage sequence (at $t = 0$ ms). A corresponding BE-PFM amplitude image is displayed in Fig. S1. The continuous change of the ferroelectric domain structure is displayed in a movie in the Supplementary Information (Supplementary Movie 1). A zoom-in into the region marked by the blue square, featuring its voltage-dependent domain-structure evolution, is displayed in **d**. The time-dependent BE-PFM phase (after the application of the DC pulses) is extracted along the dashed white line and displayed in Fig. 3d. The dimension of the square is $0.66 \times 0.66\,\mu m^2$.

structure. For example, the domain wall movements are largely suppressed in the vicinity of the vortex cores, corroborating that the vortices act as anchor points for the ferroelectric domain walls[36,37]. In contrast, substantial changes in the domain wall position are observed away from the vortex lines as presented in Fig. 1d. Figure 1d shows the voltage-dependence of the BE-PFM phase response for the area marked by the blue square in Fig. 1b, c. To demonstrate the effect of the electric field on the ferroelectric domains, the initial domain wall position is highlighted by green dots. Under the application of a negative voltage the $-P$ domains contract, followed by an expansion under a positive voltage. Most interestingly for this work, we find that the initial and final domain-wall positions in Fig. 1d are almost indistinguishable, indicating that the domain walls revert to their starting position after applying the bipolar electric-field cycle. The behavior is consistent over the sample as confirmed by Fig. S4, which displays the analysis at a different position. Our results align with previous scanning electron microscopy studies, which demonstrate that the initial and final state after ion- or electron-beam irradiation are similar[38]. In contrast to those studies, we apply voltage via an electrical contact, enabling precise monitoring of domain wall positions under controlled conditions.

To understand the local polarization reversal, we examine the voltage-dependent BE-PFM phase response on the nanoscale in the

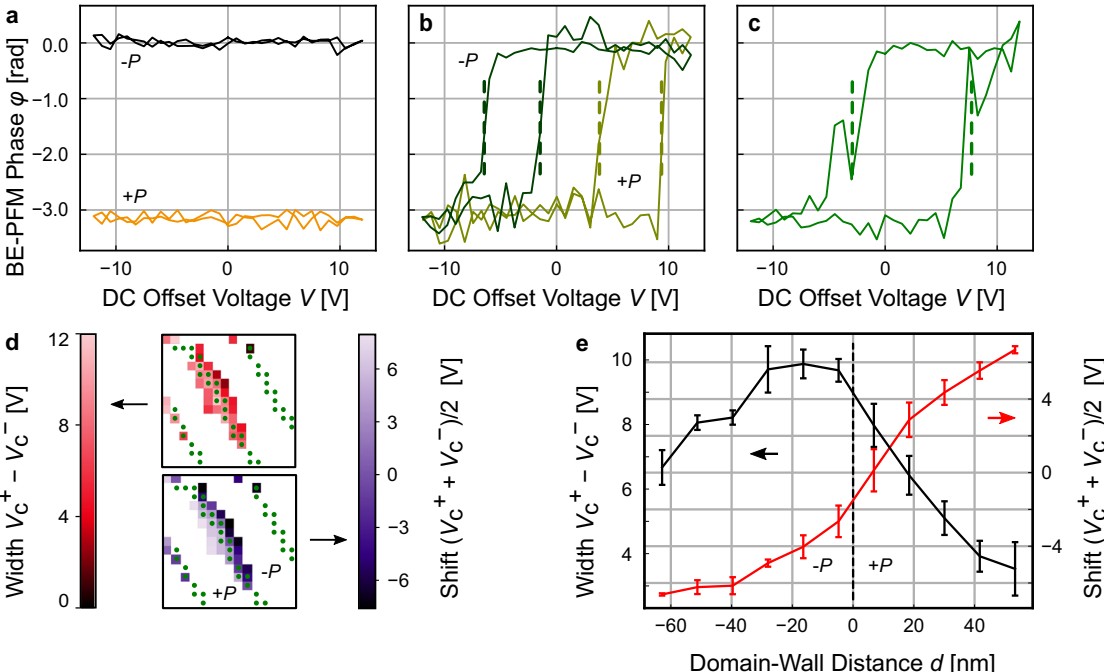

**Fig. 2 | Nanometric evaluation of the switching behavior in ErMnO₃.** The voltage-dependent BE-PFM phase response illustrates distinct switching types relative to the distance from the ferroelectric domain wall: **a** the BE-PFM phase response is constant for all DC offset voltages and no switching is observed at distances larger than 150 nm from the domain wall. The −P domains exhibit a voltage-independent phase of 0, whereas the +P domains exhibit a phase of −π. **b** Asymmetric hysteresis loop measured at two different sites of the domain wall at a distance of -30 nm. The light green loop represents the response measured in a +P domain, and the dark green loop corresponds to the response measured in −P domain. **c** Symmetric hysteresis observed at the position of the ferroelectric region of interest (Fig. 1d). domain wall. Dashed lines in **b** and **c** represent the positive and negative coercive voltage $V_c^+$ and $V_c^-$, respectively. **d** Illustration of the width and horizontal shift of the hysteresis loop, showcasing the validity of our classification (**a**–**c**) for the area of interest (Fig. 1d). Green dots indicate the position of the ferroelectric domain wall prior to the application of the bipolar voltage sequence ($t = 0$ ms). **e** Hysteresis shift and width, assessed in the vicinity of the domain wall in the section shown in Fig. 2d, depicted as a function of the distance from the domain wall position (displayed as a black dashed line) averaged between $t = 0$ ms and $t = 250$ ms. The error bars indicate the standard deviation of all pixels grouped into the corresponding data point.

region of interest (Fig. 1d). The BE-PFM phase response can be classified as a function of the distance from the domain wall. Representative responses are displayed in Fig. 2a–c. At distances exceeding 150 nm from the original domain wall position, the BE-PFM phase response is independent of the applied voltage pulse (Fig. 2a), showing that the local polarization does not switch up to the maximum applied voltage of ±12.0 V. As we approach the domain wall, highly asymmetric hysteretic BE-PFM phase responses emerge as a function of the voltage, with the effect being position-dependent, varying between 40 nm and 150 nm from the initial domain wall position. The sign of the asymmetry in the hysteresis depends on the side from which the domain wall is approached (Fig. 2b). For both sides, a change of the BE-PFM phase by π rad under the maximum applied voltage can be measured, which is an indication of local polarization reversal. Depending on the side of the domain wall, the asymmetry in the hysteresis loop is expressed by the absence of a negative or positive coercive voltage. Mechanistically, the BE-PFM data indicates that polarization reversal occurs by attraction or repulsion of the ferroelectric domain wall towards or away from the AFM tip[39,40], implying that the domain wall reverts back to its initial position after the bipolar electric-field cycle. Finally, at a position close to the ferroelectric domain wall (Fig. 2c), a hysteresis loop develops as expected for electric-field-driven polarization reversal. The behavior can be quantified by measuring the width of the hysteresis loop, $V_c^+ - V_c^-$, and its shift, $(V_c^+ - V_c^-)/2$. Voltages $V_c^+$ and $V_c^-$ are displayed as dashed lines in Fig. 2b and c. As depicted in Fig. 2d, the width of the hysteresis loop reaches up to -12 V at the domain wall and gradually decreases away from it. The hysteresis shift reaches maximum values of −6 V and +6 V away from the domain wall and gradually decreases towards the position of the wall. As expected, the hysteresis shift is

positive when approaching the domain wall from the +P domain and has a negative value when approaching the domain from the P domain. Figure 2e displays the average hysteresis width alongside the hysteresis shift plotted as a function of the distance from the domain wall position (the position is indicated by a dashed straight line). The shift of the hysteresis loop is symmetric with respect to the initial domain wall position. The asymmetry in the hysteresis width can be explained based on surface charges, which can lead to a substantial bias in ferroelectric switching[41]. The asymmetric switching behavior is consistent with previous spatially resolved measurements and plays an important role at the local scale[38]. The observed switching behavior (Fig. 2) is interesting as it shows that distinct switching behaviors can be achieved within the same electric field range at the nanoscale, depending on the relative positioning of the electrode with respect to the domain walls and vortex lines.

To evaluate whether the response to the electric field is specific to the ErMnO₃ polycrystal or representative for the family of hexagonal manganites in general, we compare the experimental results with phase-field simulations[42]. Following the established phase-field model for hexagonal manganites, the system is described by a Landau expansion of the trimerization tilt amplitude $Q$, the azimuthal tilt phase $\Phi$, and the polar mode $\mathcal{P}$[20,43,44], as elaborated in the method section. The model reproduces the characteristic domain structure of ErMnO₃ observed in our experiment (Fig. 1 and S4), including the topologically protected vortex lines, as displayed in Fig. 3a. Importantly, the data allows for investigating the impact of an electric field on different surfaces and within the bulk material, giving a three-dimensional (3D) model of the electric-field-induced domain structure. For this purpose, we consider the coupling between the energy density

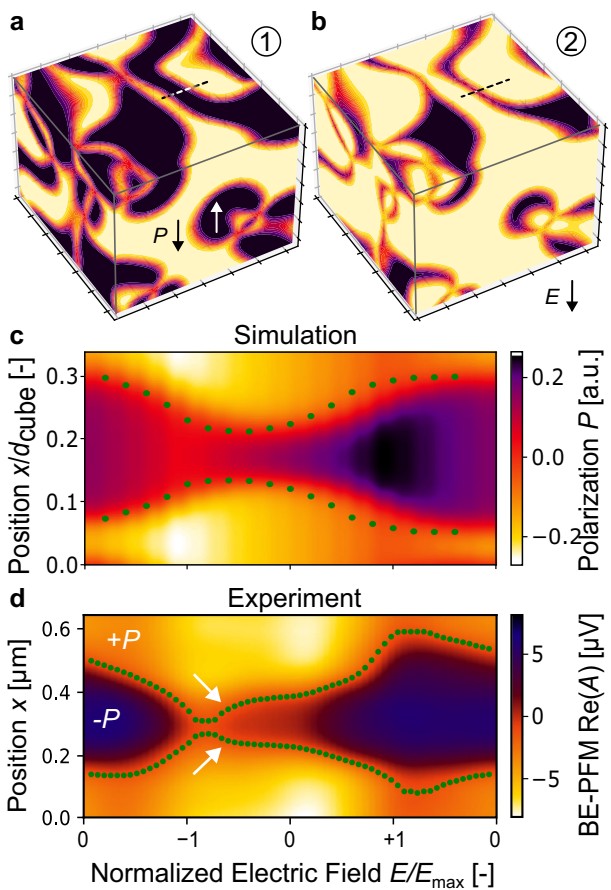

**Fig. 3 | Phase-field simulations showing the relation between the electric field and the domain structure.** The influence of an electric field on the ferroelectric domains is displayed for **a** the initial configuration without an applied electric field and **b** at the maximum negative electric field. The continuous change of the simulated ferroelectric domain structure is displayed in a movie in the Supplementary Material (Supplementary Movie 2). **c** The polarization is extracted along the dotted line in panels **a** and **b** as a function of an equivalent triangular-shaped voltage sequence, analogous to our experiment (Fig. 1a). **d** Voltage-dependent data of the BE-PFM phase, obtained from experiment, extracted along a dotted line of the area of interest in Fig. 1d is displayed. The green dots represent a guide to the eye for the electric-field-dependent positions of the domain wall, determined through thresholding. Jump-like changes in the position are highlighted by white arrows.

and the electric field as an additional term in the free energy, $F = F_{Landau} - PE$, where $P$ is the polarization and $E$ is the external electric field, as explained in more detail in the method section.

The simulated domain structure for a triangular electric field sequence analogous to the experiment (Fig. 1a) is displayed in Fig. 3a and b, visualizing the electric-field driven evolution of the ferroelectric domain structure from the initial to the maximum electric field, respectively. The simulation reproduces the experimentally observed contraction and expansion of the ferroelectric domains in response to the applied electric field. Close to the vortex lines, variations in the domain structure are much less pronounced than elsewhere, corroborating that these lines act as anchor points for the domain walls (Fig. S6a)[36,37]. Interestingly, in agreement with our experimental observations, the simulation shows that the ferroelectric domain walls return to their initial position after the bipolar electric-field cycle. Given its 3D nature, the phase-field simulations extend the reversible long-range motion observed in the BE-PFM data to the bulk of the material. The importance of the improper ferroelectric nature and the topologically protected vortex lines for the reversibility of the domain

structure is visualized in Fig. S6b and c. The observed behavior is fundamentally different from proper ferroelectric materials, which often show irreversible domain-structure changes during electric-field cycling, where the domain structure is either transformed into a mono-domain state[45] or spatially unrelated configurations[46]. The reversibility is displayed by the electric-field-dependent evolution of two representative walls in Fig. 3c. Green dots are used to indicate the positions of the ferroelectric domain walls, visualizing how they evolve as a function of the electric field. For comparison, experimental data extracted from Fig. 1d are presented in Fig. 3d. Different from the simulation, jump-like changes in position, exemplarily highlighted by white arrows, arise in the experiment, which we attribute to imperfections in the crystallographic structure of the material. For example, it has been calculated that oxygen interstitials have a lower formation energy at the domain walls and, hence, promote pinning[26,47]. Despite such jump-like changes, experiment and theory qualitatively exhibit the same behavior. The domain walls relax back to their initial positions after the bipolar electric-field cycle. This behavior suggests that the reversibility in the domain switching is linked to the improper ferroelectric order of ErMnO$_3$, with the relaxation of the primary structural order parameter acting as an additional resorting force, overcoming the pinning potential of defects. The phase-field simulations incorporate restoring forces through the stiffness term, which is directly associated with the primary structural order parameter (Eq. 1). Remarkably, the domain walls revert to their original position after having traversed distances of up to $284 \pm 25$ nm (see Fig. 1d), moving between displacements under maximum positive (+12 V) and negative voltages (−12 V). Such reversible long-range domain wall motions are consistently observed at various positions across the sample. Another example is presented in Fig. S4 (Supplementary Movie 3), showing the case of neighboring stripe-like domains, where the domain walls are displaced by about $150 \pm 25$ nm under application of the electric field as evaluated in Fig. S5. In general, even larger traveling distances may be expected if higher voltages are applied with the upper limit being constrained by the proximity of neighboring domain walls. In addition, by optimizing the synthesis and reducing the amount of lattice imperfections, the emergence of defect-related jump-like domain wall motions may be suppressed, facilitating deterministic control with even larger traveling distances. Most importantly, ErMnO$_3$ shows this reversibility in its pristine state, that is without the need for engineering by, e.g., defect dipoles[10] or point defects[12,14]. To gain additional insight into the microscopic interactions that govern the domain wall motion in ErMnO$_3$, extended atomic-level calculations are required, which could be performed as a next step. Most importantly for the present study, the agreement between the BE-PFM measurements and the phase-field simulations reflects that the observed behavior is common to improper ferroelectric hexagonal manganites and not specific to the ErMnO$_3$ polycrystal under investigation. Furthermore, this finding highlights that chemical defects play a secondary role in this system.

In conclusion, we have visualized the electric-field-driven local polarization reversal in improper ferroelectric ErMnO$_3$ using BE-PFM with nanometric precision. We attribute this phenomenon to the improper nature of the ferroelectric order, with the structural order parameter acting as a restoring force that promotes a relaxation of the domain walls to their initial position. Our observation of reversible long-range domain wall movements in ErMnO$_3$ is consistent with reports on the electric-field-induced contraction of domains into meandering lines[19,21] and extends electric-field-dependent transmission electron microscopy studies[36] to the level of ferroelectric domains. We expect that the observed reversibility of the switching process is not limited to ErMnO$_3$ and the family of hexagonal manganites. Other possible candidate materials include isostructural ferrites[48], indates[49], and gallates[50], as well as hexagonal tungsten bronzes[51] or 2H compounds[52]. To further increase the traveling distance of the domain walls or reduce the amplitude of the driving electric field, parameters such as temperature

and mechanical pressure may be leveraged to reduce the coercive field[53], representing a pathway for optimizing the dynamical responses towards potential device applications. The demonstrated controllable and reversible motion of domain walls holds promise for advanced devices such as sensors[54] and tunable capacitors[55]. For sensors, the demonstrated reversibility enables dynamic adjustment of the figures of merit, whereas in capacitors, it facilitates precise fine-tuning of the dielectric constant. In both cases, the electric field stimulates a reversible dynamic shift in domain wall position. In addition, the dynamic responses may be utilized to translate electric input signals into complex domain wall displacements with well-defined nonlinear relaxation behavior, providing all key characteristics required for reservoir computing and giving new opportunities for in-materio computing[56].

## Methods

### Synthesis and sample preparation

Synthesis of $ErMnO_3$ powder was done by a solid-state reaction of dried $Er_2O_3$ (99.9% purity; Alfa Aesar, Haverhill, MA, USA) and $Mn_2O_3$ (99.0% purity; Sigma-Aldrich, St. Louis, MO, USA) powders. The powders were mixed and ball-milled (BML 5 witeg Labortechnik GmbH, Wertheim, Germany) for 12 hrs at 205 rpm using yttria-stabilized zirconia milling balls of 5 mm diameter and ethanol as dispersion medium. The reaction to $ErMnO_3$ was done by stepwise heating to 1000 °C, 1050 °C, and 1100 °C with a dwell time of 12 hrs. More details on the powder processing can be found in ref. [28]. The powder was isostatically pressed into samples of cylindrical shape at a pressure of 200 MPa (Autoclave Engineers, Parker-Hannifin, Cleveland, OH, USA). Sintering was carried out in a closed alumina crucible at a temperature of 1350 °C for 4 hrs with a heating and cooling rate of 5 K/min (Entech Energiteknik AB; Ängelholm, Sweden).

### PFM measurements

Prior to PFM measurements, the samples were lapped with a 9-μm-grained $Al_2O_3$ water suspension (Logitech Ltd., Glasgow, UK) and polished using silica slurry (SF1 Polishing Fluid, Logitech AS, Glasgow, Scotland). BE-PFM measurements were conducted with the same Cr/Pt coated Multi75E-G AFM probe (force constant, $k \approx 3$ N/m, Budget Sensors, Sofia, Bulgaria), using a Cypher atomic force microscope (Oxford Instrument, Abingdon, UK). For BE-PFM imaging the electrical bias was applied to the surface via the probe tip, while the sample was grounded. More information on the measurement technique are provided in ref. [32]. The tip nominal radius was ~25 nm and the pixel size was ~47 nm. For the analysis of the data in Figs. [2] and [3], we considered that the regular grid structure of the scan is non-collinear with the features of interest under investigation. BE-PFM was achieved by arbitrary-wave generator and data-acquisition electronics. A custom software program was employed to generate the probing signal and record local amplitude and phase hysteresis loops. BE-PFM measurements were performed at a frequency range of 360 to 420 kHz, a power spectral density of 0.57 $V^2$/kHz, and an amplitude of 5 V, respectively. Simple harmonic oscillator fits were subsequently applied to the measured spectra to extract amplitude and phase information at the resonance frequency. The SS-PFM triangular-square function had a maximum of ±12 V, applied over 600 ms with a sequence of 64 steps of on-field and off-field measurements.

### Phase-field simulations

Phase-field simulations were performed based on the Landau expansion of the free energy of ferroelectric hexagonal manganites as described in ref. [20] as

$$F_{Landau}(Q, \Phi, \mathcal{P}) = \frac{a}{2}Q^2 + \frac{b}{4}Q^4 + \frac{Q^6}{6}(c + c' \cos 6\Phi) - gQ^3\mathcal{P}\cos 3\Phi - \frac{g'}{2}Q^2\mathcal{P}^2$$
$$+ \frac{a_P}{2}\mathcal{P}^2 + \frac{1}{2}\sum_{i=x,y,z}[s_Q^i(\partial_i Q \partial_i Q + Q^2 \partial_i \Phi \partial_i \Phi) + s_P^i \partial_i \mathcal{P} \partial_i \mathcal{P}], \quad (1)$$

where $Q$ is the amplitude and $\phi$ is the angle of the lattice-trimerizing bipyramidal $MnO_5$ tilt, and $\mathcal{P}$ is a displacement field that corresponds to the polar mode $\Gamma_2^-$. This displacement field $\mathcal{P}$ is proportional to the polarization, $P$, of the system. To describe the coupling between the polarization, $P$, and the external electric field, $E$, an additional term $-PE$ is included. The term $s_Q^i(\partial_i Q \partial_i Q + Q^2 \partial_i \Phi \partial_i \Phi)$ constitutes the stiffness term representing the structural order parameter and accounting for inhomogeneities in the structural order. The structural order is coupled to the ferroelectric order through the terms $(-gQ^3\mathcal{P}\cos 3\Phi - \frac{g'}{2}Q^2\mathcal{P}^2)$. The parameters for the Landau expansion were chosen as in ref. [20], as $a = -2.626$ eVA$^{-2}$, $b = 3.375$ eVA$^{-4}$, $c = 0.117$ eVA$^{-6}$, $c' = 0.108$ eVA$^{-6}$, $a_P = 0.866$ eVA$^{-2}$, $g = 1.945$ eVA$^{-4}$, $g' = 9.931$ eVA$^{-4}$, $s_Q^z = 15.40$ eV, $s_Q^x = 5.14$ eV, and $s_P^z = 52.70$ eV. To ensure stability, the gradient energy coefficient was set to $s_P^x = +8.88$ eV as validated in ref. [44]. Our system was simulated with a uniform Cartesian computational mesh with spacing $d_x = d_y = 0.2$ nm and $d_z = 0.3$ nm. Our simulations were performed on a mesh with size $n_x = n_y = n_z = 64$. Periodic boundary conditions were chosen in all three principal directions to simulate a bulk system. The Ginzburg-Landau equations were integrated with a Runge-Kutta 4 integrator with time steps of $\Delta t = 5 \cdot 10^{-4}$. The system was initialized with random values of all three order parameters. The initial domain pattern was then generated by evolving the system for $8 \cdot 10^4$ time steps, with no external electric field applied, i.e., $E = 0$. From this initial domain pattern, the system was integrated with an external field applied parallel to the polarization direction, i.e.,

$$E(n) = \begin{cases} \frac{4n}{N} \cdot E_{max} & \frac{n}{N} \leq 1/4 \\ \left(1 - 4\left(\frac{n}{N} - \frac{1}{4}\right)\right) \cdot E_{max} & \frac{1}{4} < \frac{n}{N} \leq \frac{3}{4} \\ \left(-1 + 4\left(\frac{n}{N} - \frac{3}{4}\right)\right) \cdot E_{max} & \frac{3}{4} < \frac{n}{N} \end{cases} \quad (2)$$

which is similar to the experimentally applied bipolar voltage sequence displayed in Fig. [1]a, yet with a flipped sign of $E(n)$. Here, $N = 2 \cdot 10^4$ is the number of the time steps of a full cycle and $n$ is a given time step of the cycle. The maximal electric field has been chosen to $E_{max} = 0.11$ V/Å.

## Data availability
The data that support the findings of this study are available from the corresponding author upon reasonable request.

## Code availability
The applied data evaluation algorithms applied are available at https://zenodo.org/records/14624858. Further data evaluation analysis that supports the findings of this study are available from the corresponding author upon reasonable request.

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

## Acknowledgements

J.S. acknowledges inspiring discussions with D. Damjanovic. O. W. Sandvik is acknowledged for initial analysis of the data. M.Z.

acknowledges funding from the Studienstiftung des Deutschen Volkes via a doctoral grant and the State of Bavaria via a Marianne-Plehn scholarship. J.S. and D.M. acknowledge NTNU Nano for the support through the NTNU Nano Impact Fund and funding from the European Research Council (ERC) under the European Union's Horizon 2020 Research and Innovation Program (Grant Agreement No. 863691). J.S. acknowledges the support of the Alexander von Humboldt Foundation through a Feodor-Lynen research fellowship and the German Academic Exchange Service (DAAD) for a Post-Doctoral Fellowship (Short-term program). D.M. thanks NTNU for support through the Onsager Fellowship Program and the Outstanding Academic Fellow Program. A.M.M. acknowledges funding from the Swiss National Science Foundation (SNSF) through grant numbers 200021_178825 and 200021_215423. M.Z. and I.K. acknowledge funding by the Deutsche Forschungsgemeinschaft (DFG, German Research Foundation), TRR 360, grant number 492547816. S.V.K. acknowledges support by the Center for Advanced Materials and Manufacturing (CAMM), the NSF MRSEC center. The scanning probe microscopy research was supported by the Center for Nanophase Materials Sciences (CNMS), which is a US Department of Energy, Office of Science User Facility at Oak Ridge National Laboratory.

## Author contributions

M.Z.: Software (Data Analysis), Validation, Formal Analysis, Visualization. A.M.M.: Software (Phase-Field Simulations), Formal Analysis. K.P.K.: Investigation, S.M.N.: Validation, S.V.K.: Methodology. I.K.: Supervision, M.F. Writing – Review & Editing. Th.L.: Supervision. N.D.: Methodology, Supervision. D.M.: Conceptualization, Funding Acquisition, Writing – Review & Editing, Supervision. J.S.: Conceptualization, Funding Acquisition, Writing – Original Draft, Project Administration, Supervision.

## Funding

 Olavs Hospital - Trondheim University Hospital).

## Competing interests

The authors declare no competing interests.

## Inclusion and Ethics Statement

This study adheres to ethical research practices, ensuring inclusivity and diversity in its scope and collaboration. The research team consists of members from diverse backgrounds, contributing varied perspectives and expertise. No individual or group was excluded based on ethnicity, gender, or other characteristics. The work complies with all relevant ethical guidelines and institutional policies.
