## [Transparent Peer Review file · Nature Communications]

Reversible long-range domain wall motion in an improper ferroelectric

Corresponding Author: Dr Jan Schultheiß

Version 0:

Reviewer comments:

Reviewer #1

(Remarks to the Author)

The authors studied the electric-field-driven local polarization reversal in improper ferroelectric ErMnO₃ using BE-PFM with nanometric precision. The paper is well organized with sufficient references. It will be useful for researchers in the field of ferroelectric domains and other fields as well. Due to the interest for the topic and the relevant amount of new results presented in the manuscript, I recommend it for publication. Here are my comments about the manuscript:

- 1) The authors mentioned that domain walls can recover to the initial position after repeated application of electric field. Ensuring the stability and controllability of domain wall motion is a crucial challenge in device design. Whether the impact of external factors, such as temperature and pressure, on the stability of domain wall motion is taken into account
- 2) The authors point out that the reversible motion of domain walls under the action of an electric field is facilitated by the improper ferroelectric order, with the structural order parameter acting as a restoring force. Have you further explored how this restoring force influences the motion of domain walls?
- 3) During the experimental process of this work, there were jump-like changes in position, attributed to imperfections in the crystalline structure of the material. Will this defect potentially compromise the accuracy of the experimental results? Did the authors take any measures to mitigate such impacts?

Reviewer #2

(Remarks to the Author)

The paper investigates domain wall dynamics in the geometrically driven ferroelectric ErMnO₃. Band-excitation piezoresponse force microscopy is employed to track domain wall movement, with measurements indicating that, when situated away from topologically protected structural vortex lines, domain walls tend to return to their original positions after bipolar electric field cycling, even after displacements of around 280 nm. Complementary phase field simulations suggest this phenomenon is characteristic of hexagonal manganites and can be attributed to their improper ferroelectric order, with the structural order parameter encouraging the relaxation of displaced domain walls and facilitating their return to original configurations. The experimental methodology is well described. The findings are not only original and significant but also presented in a manner that underscores their potential impact on the broader field of ferroelectric materials research. Given the manuscript's quality, I recommend it for publication in Nature Communications in its current form.

Reviewer #3

(Remarks to the Author)

Thanks for the opportunity to review this work.

Using band-excitation piezoresponse force microscopy with nanoscale spatial accuracy, the author demonstrated the reversible domain wall motion of ErMnO₃ under cycling electric field, and with the help of phase field simulation, they verified its correlation with improper ferroelectric order and topological vortex lines.

However, I have some questions:

1. According to my understanding, the authors presented the reversible changes of domain walls in a small range (~250nm) in case that the topological network structure of the sample remains basically unchanged (as mentioned in lines 235-236, 'limited by the proximity of adjacent domain walls'). Even more, the positions of the domain walls were not exactly reversible with a certain degree of randomness. Thus, the significance for the practical applications needs to be further explained by

the authors. For example, at the end of the abstract, the authors mentioned 'tunable capacitors or sensors'. Do they have more detailed explanations and supporting materials?

2. We know that the state of a system is determined by the minimum energy. In this manuscript, the phase field simulation started from an initial state (of course with low energy), and then went through a series of intermediate processes similar to disturbances. Coming back to the original conditions, the system's state reversibly returned, which can be expected regardless of the material structure. Therefore, the authors need to clarify the uniqueness of this phase field simulation.

3. In lines 106-107, they mentioned 'near the structural vortex lines, we do not resolve domain wall movements'. However, the red arrow in Figure 1b points to a white area, while the same area in Figure 1c turns black. Does it indicate the domain wall movement near the structural vortex lines?

4. The explanation of Figure 2e was insufficient in the text. For example, (1) why the curves are not symmetric? and (2) why 15 nm is a turning point?

5. In lines 26-29, they mentioned '...the reversible long-range motion is intrinsic to the hexagonal manganites, linking it to their improper ferroelectricity and topologically protected structural vortex lines, which serve as anchor point for the ferroelectric domain walls'. Now that the importance of vortex lines was emphasized in the text, why didn't they provide a comparison between the simulation and the experiment near a vortex line also in Figure 3? In addition, the domain configuration simulated in Figure 3 differs significantly from that in Figures 1 and S4. Why not adopt a domain configuration similar to the experiment?

6. In lines 230-231, they mentioned '...with the relaxation of the primary structural order parameter acting as an additional resorting force'. Where was this point displayed? Or, were there any related terms in the calculation formula of phase field simulation?

7. In lines 234-235, they mentioned '...even larger distances may be expected if higher voltages are applied...'. What is the maximum voltage and distance that can be achieved for the samples in the manuscript?

Version 1:

Reviewer comments:

Reviewer #1

(Remarks to the Author)

I find the authors have addressed all the concerns raised by the reviewer, and now I recommend this paper for publication.

Reviewer #3

(Remarks to the Author)

The authors have addressed all my concerns. Thanks for their explanations.
I recommend this manuscript for publication.

Point-by-point response for manuscript NCOMMS-24-69136

Reply to the comments of Reviewer #1

The authors studied the electric-field-driven local polarization reversal in improper ferroelectric ErMnO₃ using BE-PFM with nanometric precision. The paper is well organized with sufficient references. It will be useful for researchers in the field of ferroelectric domains and other fields as well. Due to the interest for the topic and the relevant amount of new results presented in the manuscript, I recommend it for publication. Here are my comments about the manuscript:

We greatly appreciate the reviewer's encouraging remarks and the recognition of the relevance and quality of our study on electric-field-driven local polarization reversal in improper ferroelectric ErMnO₃. We have thoroughly considered the reviewer's constructive comments and have addressed them to further improve our manuscript.

Comment #1:

The authors mentioned that domain walls can recover to the initial position after repeated application of electric field. Ensuring the stability and controllability of domain wall motion is a crucial challenge in device design. Whether the impact of external factors, such as temperature and pressure, on the stability of domain wall motion is taken into account.

Response #1:

In the investigated voltage regime ($-12 \text{ V} < U < 12 \text{ V}$), we consistently observe that domain walls can move reversibly by up to approximately 250 nm at room temperature. Slight variations in the amplitudes of motion will naturally occur under temperature or pressure variations, as these parameters co-determine the material's coercive field (<https://doi.org/10.1063/1.5026732>). In general, any reduction of the coercive field is expected to increase the travel distance. Our findings demonstrate that reversible long-range movements are fundamentally possible, whereas quantitative details regarding the impact of mechanical pressure and other influencing factors remain to be studied; a corresponding note has been added to the conclusion of the manuscript (page 12).

To further increase the traveling distance of the domain walls or reduce the amplitude of the driving electric field, parameters such as temperature and mechanical pressure may be leveraged to reduce the coercive field,^[54] representing a pathway for optimizing the dynamical responses towards potential device applications.

Comment #2:

The authors point out that the reversible motion of domain walls under the action of an electric field is facilitated by the improper ferroelectric order, with the structural order parameter acting as a restoring force. Have you further explored how this restoring force influences the motion of domain walls?

Response #2:

In hexagonal manganites, the electric field acts on the secondary order parameter (improper ferroelectric polarization). This secondary order parameter is coupled to the primary structural

order parameter that determines the domain structure (<https://doi.org/10.1038/nmat3786>). As such, the primary structural order parameter gives the restoring force as reflected by our phase field simulations. The simulations, which incorporate the stiffness term $s_Q^i(\partial_i Q \partial_i Q + Q^2 \partial_i \Phi \partial_i \Phi)$, reveal that inhomogeneities in the structural order generate restoring forces through electric-field-induced confinement of the ferroelectric domain structure. To clarify this point, we have added a sentence on page 10 of the manuscript.

The phase-field simulations incorporate restoring forces through the stiffness term, which is directly associated with the primary structural order parameter (eq. 1).

To further clarify the meaning of the respective terms in the phase-field simulations, we have extended in the respective part of the methodology:

To describe the coupling between the polarization, P , and the external electric field, E , an additional term $-PE$ is included. The term $s_Q^i(\partial_i Q \partial_i Q + Q^2 \partial_i \Phi \partial_i \Phi)$ constitutes the stiffness term, representing the structural order parameter and accounting for inhomogeneities in the structural order. The structural order is coupled to P through the terms $(-gQ^3P \cos 3\Phi - \frac{g'}{2} Q^2 P^2)$.

Comment #3:

During the experimental process of this work, there were jump-like changes in position, attributed to imperfections in the crystalline structure of the material. Will this defect potentially compromise the accuracy of the experimental results? Did the authors take any measures to mitigate such impacts?

Response #3:

The reviewer is correct that imperfections in the crystalline structure can influence the domain wall motion. Such defects may impact the field dependence of the motion and potentially limit the traveling distance. The values are thus to be considered a lower limit for the traveling distance, whereas even larger distances can be expected for an ideal defect-free system. A discussion on the influence of lattice imperfections has been added on page 10 of the manuscript.

In addition, by optimizing the synthesis and reducing the amount of lattice imperfections, the emergence of defect-related jump-like domain wall motions may be suppressed, facilitating deterministic control with even larger traveling distances.

Regarding the accuracy of our experimental results, we are aware that defects might play a role, and we have therefore conducted measurements at multiple positions as shown in Figure S4. At the second position (Figure R1, included as Figure S5), we examined the electric-field-dependency of two domains and quantified the traveling distance (150 ± 25 nm), demonstrating that the reversible domain wall motion over long distances is a general phenomenon. Most importantly, the additional data corroborates that the mechanism is independent of the position and not determined by local imperfections. Furthermore, the investigation of two neighboring domains (Figure R1) reveals that the reversible domain wall movement is robust against crosstalk between domains. A corresponding sentence has been added on page 10 of the manuscript.

Such reversible long-range domain wall motions are consistently observed at various positions across the sample. Another example is presented in Figure S4, showing the case of neighboring stripe-like domains, where the domain walls are displaced by about 150 ± 25 nm under application of the electric field as evaluated in Figure S5.

Figure R1. Local influence of the electric field on stripe-like ferroelectric domains. Voltage-dependent data from the experiment are shown as BE-PFM phase extracted along the dotted line in Fig. S4. The green dots serve as a visual guide, indicating the electric-field-dependent positions of the domain wall, determined through thresholding. The walls traverse distances up to 150 ± 25 nm, returning to their original positions, moving between maximum positive and negative voltages. No crosstalk is observed between the neighboring stripe-like domains that would influence the reversible domain wall movement.

Reply to the comments of Reviewer #2

The paper investigates domain wall dynamics in the geometrically driven ferroelectric ErMnO_3 . Band-excitation piezoresponse force microscopy is employed to track domain wall movement, with measurements indicating that, when situated away from topologically protected structural vortex lines, domain walls tend to return to their original positions after bipolar electric field cycling, even after displacements of around 280 nm. Complementary phase field simulations suggest this phenomenon is characteristic of hexagonal manganites and can be attributed to their improper ferroelectric order, with the structural order parameter encouraging the relaxation of displaced domain walls and facilitating their return to original configurations. The experimental methodology is well described. The findings are not only original and significant but also presented in a manner that underscores their potential impact on the broader field of ferroelectric materials research. Given the manuscript's quality, I recommend it for publication in Nature Communications in its current form.

We sincerely thank the reviewer for reading the manuscript and the kind and supportive feedback. We are delighted that our work on domain wall dynamics in improper ferroelectric ErMnO_3 was well received and appreciate the recommendation for publication.

Reply to the comments of Reviewer #3

Thanks for the opportunity to review this work.

Using band-excitation piezoresponse force microscopy with nanoscale spatial accuracy, the author demonstrated the reversible domain wall motion of ErMnO₃ under cycling electric field, and with the help of phase field simulation, they verified its correlation with improper ferroelectric order and topological vortex lines.

However, I have some questions:

We thank the reviewer for her/his thorough evaluation and insightful comments on our manuscript. Her/his acknowledgment of our work is greatly appreciated. We have addressed the reviewer's remaining questions and incorporated the necessary revisions as detailed below.

Comment #1:

According to my understanding, the authors presented the reversible changes of domain walls in a small range (~250nm) in case that the topological network structure of the sample remains basically unchanged (as mentioned in lines 235-236, 'limited by the proximity of adjacent domain walls'). Even more, the positions of the domain walls were not exactly reversible with a certain degree of randomness. Thus, the significance for the practical applications needs to be further explained by the authors. For example, at the end of the abstract, the authors mentioned 'tunable capacitors or sensors'. Do they have more detailed explanations and supporting materials?

Response #1:

The most important aspect of our work is the demonstration of reversible long-range domain-wall motion in the pristine state, i.e., without the need for sophisticated engineering steps as applied in previous studies. This finding reveals a fundamentally new intrinsic mechanism that leads to unusual domain wall dynamics. Aside from the intriguing physics and the unusual material's response, the intrinsic nature of this mechanism is compelling for device applications that leverage domain wall motions, even if there is locally some minor degree of randomness. Notably, the dimensions of read-out electrodes in devices, typically exceeding 10 nm when fabricated using photolithography (<https://doi.org/10.1002/adom.201900598>), are larger than the positional uncertainty observed in our experiment. To elaborate on this aspect, we added a paragraph and references 55-57 on page 12 of the manuscript.

The demonstrated controllable and reversible motion of domain walls holds promise for advanced devices such as programmable sensors^[55] and tunable capacitors^[56]. For sensors, the demonstrated reversibility enables dynamic adjustment of the figures of merits, whereas in capacitors, it facilitates precise fine-tuning of the dielectric constant. In both cases, the electric field stimulates a reversible dynamic shift in domain wall position. In addition, the dynamic responses may be utilized to translate electric input signals into complex domain wall displacements with well-defined nonlinear relaxation behavior, providing all key characteristics required for reservoir computing and giving new opportunities for in-materio computing.^[57]

Comment #2:

We know that the state of a system is determined by the minimum energy. In this manuscript, the phase field simulation started from an initial state (of course with low energy), and then went through a series of intermediate processes similar to disturbances. Coming back to the original conditions, the system's state reversibly returned, which can be expected regardless of the material structure. Therefore, the authors need to clarify the uniqueness of this phase field simulation.

Response #2:

Different from what the reviewer assumes, the return of the domain walls to the initial state in the phase field simulations is not a simple consequence of the initial relaxation of the system to a low energy configuration. This is best illustrated by analyzing the simulated domain structure before and after applying an electric field to a domain, evaluated along the dashed line shown in Fig. R2 (included as Figure S6). We find subtle variations in the domain structure when comparing the initial (1) and final (4) state. Such variations in the domain structure are important as they corroborate that the system does not just return to the original domain structure but has the degree of freedom to acquire a different low energy state. With this additional analysis, we can thus safely exclude that the system simply goes “through a series of intermediate processes similar to disturbance”; in contrast, every electric field cycle induces minor local deviations, consistent with our experimental observations. Furthermore, in full agreement to the pinning effect associated with the vortex cores, differences between the initial (1) and final (4) state are more pronounced further away from vortex cores (rising edge in Fig. R2b).

Figure R2. Repeated domain switching in phase field simulation. (a) Exemplary domain structure in the phase field simulation before applying the triangular electric field sweep cycle. For comparison of the initial and final state, the polarization along the dashed line is extracted. (b) Comparison of the ferroelectric polarization in the initial (1) and final (4) state.

We also note that the observed reversibility in our simulations (Fig. 3c) is unique, as it contrasts fundamentally with phase-field simulations of proper ferroelectrics, such as PbTiO_3 (10.1016/j.actamat.2003.10.011), where domain structures typically do not exhibit reversible switching and often transfer into a mono-domain state or irreversibly reformed under electric field cycling (10.2320/matertrans.MC200806). We have added a sentence on page 9 of the manuscript.

The importance of the improper ferroelectric nature and the topologically protected vortex lines for the reversibility of the domain structure is visualized in Fig. S6b and c. The observed behavior

is fundamentally different from proper ferroelectric materials, which often show irreversible domain-structure changes during electric-field cycling, where the domain structure is either transformed into a mono-domain state^[45] or spatially unrelated configurations.^[46]

Comment #3:

In lines 106-107, they mentioned 'near the structural vortex lines, we do not resolve domain wall movements'. However, the red arrow in Figure 1b points to a white area, while the same area in Figure 1c turns black. Does it indicate the domain wall movement near the structural vortex lines?

Response #3:

It is correct that there is domain wall movement in the vicinity of the vortex lines; however, it is substantially smaller than the movement observed far from the vortex lines (please refer to #5 for details). We have updated the text accordingly.

For example, the domain wall movements are largely suppressed in the vicinity of the vortex cores, corroborating that the vortices act as anchor points for the ferroelectric domain walls.^[36,37]

Comment #4:

The explanation of Figure 2e was insufficient in the text. For example, (1) why the curves are not symmetric? and (2) why 15 nm is a turning point?

Response #4:

We thank the reviewer for pointing this out and have extended the explanation. Consistent with literature (<https://doi.org/10.1063/1.4902396>), we attribute the asymmetry in the hysteresis width to surface charges. It is established that surface charges bias the electrical switching on the local scale, leading to pronounced asymmetries. The latter was also observed in non-contact experiments, when switching the domains in hexagonal manganites with an electron beam (<https://doi.org/10.1063/5.0038909>). In order to say more about the changes in shape, we reanalyzed the data accounting for larger distances away from the walls. From the new data, it becomes clear the shift of the hysteresis loop is symmetric with respect to the domain wall. We added a comment on page 7 of the manuscript:

The shift of the hysteresis loop is symmetric with respect to the initial domain wall position. The asymmetry in the hysteresis width can be explained based on surface charges, which can lead to a substantial bias in ferroelectric switching^[41]. The asymmetric switching behavior is consistent with previous spatially resolved measurements and plays an important role at the local scale.^[38]

Comment #5:

In lines 26-29, they mentioned '...the reversible long-range motion is intrinsic to the hexagonal manganites, linking it to their improper ferroelectricity and topologically protected structural vortex lines, which serve as anchor point for the ferroelectric domain walls'. Now that the importance of vortex lines was emphasized in the text, why didn't they provide a comparison between the simulation and the experiment near a vortex line also in Figure 3? In addition, the domain configuration simulated in Figure 3 differs significantly from that in Figures 1 and S4. Why not adopt a domain configuration similar to the experiment?

Response #5:

Motivated by the reviewer's comment, we now present the maximum changes in the ferroelectric domain structure, illustrating the positions traversed by the ferroelectric domain walls during the electric-field cycle as determined from phase-field simulations (Fig. R3, included as Figure S6a). Fig. R3 provides a direct comparison of the traveling distances in the vicinity and away from the vortex lines. It is clear from the data that the movement of a wall correlates with the distance to other domain walls. For larger distances, the electric-field driven displacement is more pronounced, implying that the movement is strongly suppressed as we approach the center of the vortices, where the domain walls meet and their distance approaches zero.

Fig. R3. Visualization of domain-wall-motion amplitude in phase-field simulations. Depicted is the maximum change in polarization for each spatial point over the entire electric-field sweep cycle. Spatial positions where this value is large indicate that the domain wall has passed through these positions during the electric-field sweep cycle. An exemplary domain wall with fixed ends in two different vertices is highlighted by a dashed line. Largest domain wall motion amplitude is observed in the middle of the line while, i.e., furthest away from the vertices, while the motion decreases in areas of high domain wall density and vanishes at the vertices.

To answer the second question of the reviewer, we note that the simulated domain structure includes all key elements to evaluate the electric field's impact on the experimental ferroelectric domain configuration. In addition to this, since our experimental observations are limited to the surface, the 3D insights from phase field simulations strengthen our argumentation that the reversibility of domain wall movement extends to the bulk of ErMnO_3 . Two clarifying statements have been added to the manuscript (pages 8 and 9).

The model reproduces the characteristic domain structure of ErMnO_3 observed in our experiment (Figures 1 and S4), including the topologically protected vortex lines, as displayed in Figure 3a.

Given its three-dimensional nature, the phase field simulations extend the reversible long-range motion observed in the BE-PFM data to the bulk of the material.

Comment #6:

In lines 230-231, they mentioned ‘...with the relaxation of the primary structural order parameter acting as an additional resorting force’. Where was this point displayed? Or, were there any related terms in the calculation formula of phase field simulation?

Response #6:

Yes, there are related terms in the calculation formula of the phase-field simulation (eq. 1). Specifically, the term $s_Q^i(\partial_i Q \partial_i Q + Q^2 \partial_i \Phi \partial_i \Phi)$, referred to as the stiffness term, accounts for inhomogeneities in the structural order. Particularly, when the domain walls are approaching each other driven by an electric field, the structural term facilitates the transition back to their original position. Furthermore, since the structural order is coupled to the ferroelectric order through the terms $(-gQ^3 \mathcal{P} \cos 3\Phi - \frac{g'}{2} Q^2 \mathcal{P}^2)$, changes in the structural order naturally induce corresponding changes in the ferroelectric order parameter (<https://doi.org/10.1038/nmat3786>). This coupling highlights how the relaxation of the primary structural order parameter contributes as an additional restoring force as mentioned in lines 230–231. To clarify that this point is explicitly addressed in the phase-field simulation, we have added a statement to the manuscript (page 10).

The phase-field simulations incorporate restoring forces through the stiffness term, which is directly associated with the structural order parameter (eq. 1).

To clarify the meaning of the respective terms in the phase-field simulations, we now extend the respective part of the methodology:

To describe the coupling between the polarization, P , and the external electric field, E , an additional term $-PE$ is included. The term $s_Q^i(\partial_i Q \partial_i Q + Q^2 \partial_i \Phi \partial_i \Phi)$ constitutes the stiffness term, representing the structural order parameter and accounting for inhomogeneities in the structural order. The structural order is coupled to P through the terms $(-gQ^3 \mathcal{P} \cos 3\Phi - \frac{g'}{2} Q^2 \mathcal{P}^2)$.

Comment #7

In lines 234-235, they mentioned ‘...even larger distances may be expected if higher voltages are applied...’. What is the maximum voltage and distance that can be achieved for the samples in the manuscript?

Response #7:

The maximum voltage applied to the sample in the manuscript is ± 12 V, resulting in a maximum travel distance of 284 ± 25 nm within the investigated region. We have included the maximum positive and negative voltage in the manuscript on page 10.

Remarkably, the domain walls revert to their original position after having traversed distances of up to 284 ± 25 nm (see Figure 1d), moving between displacements under maximum positive (+12 V) and negative voltage (−12 V).